# Effect of Europium Substitution on the Structural, Magnetic and Relaxivity Properties of Mn-Zn Ferrite Nanoparticles: A Dual-Mode MRI Contrast-Agent Candidate

**DOI:** 10.3390/nano13020331

**Published:** 2023-01-12

**Authors:** Hamidreza Saeidi, Morteza Mozaffari, Serhat Ilbey, Silvio Dutz, Diana Zahn, Gholamhassan Azimi, Michael Bock

**Affiliations:** 1Faculty of Physics, University of Isfahan, Isfahan 8174673441, Iran; 2Department of Radiology, Medical Physics, Medical Center University of Freiburg, Faculty of Medicine, University of Freiburg, Killianstr. 5a, 79106 Freiburg, Germany; 3Institute of Biomedical Engineering and Informatics, Technische Universität Ilmenau, Gustav-Kirchhof-Straße 2, 98693 Ilmenau, Germany; 4Department of Chemistry, University of Isfahan, Isfahan 8174673441, Iran

**Keywords:** MRI contrast agents, europium-substituted Mn-Zn ferrites, MRI relaxometry

## Abstract

Magnetic nanoparticles (MNPs) have been widely applied as magnetic resonance imaging (MRI) contrast agents. MNPs offer significant contrast improvements in MRI through their tunable relaxivities, but to apply them as clinical contrast agents effectively, they should exhibit a high saturation magnetization, good colloidal stability and sufficient biocompatibility. In this work, we present a detailed description of the synthesis and the characterizations of europium-substituted Mn–Zn ferrite (Mn_0.6_Zn_0.4_Eu*_x_*Fe_2−*x*_O_4_, *x* = 0.00, 0.02, 0.04, 0.06, 0.08, 0.10, and 0.15, herein named MZF for *x* = 0.00 and EuMZF for others). MNPs were synthesized by the coprecipitation method and subsequent hydrothermal treatment, coated with citric acid (CA) or pluronic F127 (PF-127) and finally characterized by X-ray Diffraction (XRD), Inductively Coupled Plasma (ICP), Vibrating Sample Magnetometry (VSM), Fourier-Transform Infrared (FTIR), Dynamic Light Scattering (DLS) and MRI Relaxometry at 3T methods. The XRD studies revealed that all main diffraction peaks are matched with the spinel structure very well, so they are nearly single phase. Furthermore, XRD study showed that, although there are no significant changes in lattice constants, crystallite sizes are affected by europium substitution significantly. Room-temperature magnetometry showed that, in addition to coercivity, both saturation and remnant magnetizations decrease with increasing europium substitution and coating with pluronic F127. FTIR study confirmed the presence of citric acid and poloxamer (pluronic F127) coatings on the surface of the nanoparticles. Relaxometry measurements illustrated that, although the europium-free sample is an excellent negative contrast agent with a high *r*_2_ relaxivity, it does not show a positive contrast enhancement as the concentration of nanoparticles increases. By increasing the europium to *x* = 0.15, *r*_1_ relaxivity increased significantly. On the contrary, europium substitution decreased *r*_2_ relaxivity due to a reduction in saturation magnetization. The ratio of *r*_2_/*r*_1_ decreased from 152 for the europium-free sample to 11.2 for *x* = 0.15, which indicates that Mn_0.6_Zn_0.4_Eu_0.15_Fe_1.85_O_4_ is a suitable candidate for dual-mode MRI contrast agent potentially. The samples with citric acid coating had higher *r*_1_ and lower *r*_2_ relaxivities than those of pluronic F127-coated samples.

## 1. Introduction

Due to their unique properties, magnetic nanoparticles (MNPs) have been widely used in a wide range of applications in different industries (e.g., magnetic cores, sensors) [1,2,3,4,5,6,7] and biomedicine (MRI, targeted drug delivery, hyperthermia, etc.) [8,9,10,11,12]. MNPs can be synthesized with a size comparable to biological molecules, such as proteins and nucleic acids, and they can be equipped with excellent physical and chemical properties like superior magnetic and optical properties, superior electrical resistivity, low eddy current and dielectric losses [13,14]. Among these materials, over the past few decades Mn–Zn ferrites (MZFs) have attracted much attention. The magnetic properties of the MZFs are strongly dependent on the phase purity, microstructure, morphology (size, shape and particle size distribution) and the types and amount of dopants [15]. MZFs have diverse biotechnical applications, such as cancer cell diagnosis [16], drug delivery [17], tumor imaging and the treatment of cancer cells [18]. Superparamagnetic MZF nanoparticles can increase water proton relaxation rates in tissue, which makes them promising candidates for MRI contrast agents [19,20]. 

In general, two different types of MRI contrast agents exist. Positive ones (e.g., based on gadolinium chelates) predominantly shorten the longitudinal relaxation time *T*_1_, so that the signal intensity in the region of interest is increased (hyperintensity in *T*_1_-weighted images) [21,22]. Negative contrast agents (e.g., magnetic iron oxides) shorten the transverse relaxation time T_2_, leading to decreased signal intensity (hypointensity in T_2_-weighted images) [23]. However, single-mode contrast agents do not always provide the required contrast enhancement for disease detection, so dual-mode contrast agents with both negative and positive contrast abilities are needed to improve accuracy and reliability [24]. To design dual-mode contrast agents, different methods have been developed recently, for example, ultrafine magnetic iron oxides, core–shell nanostructures with T_2_-weighted contrast agents as cores and T_1_-weighted contrast agents as shells, and doping paramagnetic ions into iron-based contrast agents [25]. Rare-earth ions (e.g., Sm^3+^, Eu^3+^, Gd^3+^) with large ionic radii in comparison to Fe^3+^ ions can alter the structural and physical properties of ferrites. Among them, paramagnetic ions of gadolinium (Gd^3+^ with seven unpaired electron spins) or europium (Eu^3+^ with six unpaired electron spins) offer a large magnetic moment that can alter the magnetic properties of ferrite to enhance T_1_-shortening. Thus, the doping of MZFs with these rare-earth ions may provide an alternative to ordinary dual-mode contrast agents [24,26,27]. 

For biomedical applications, these nanoparticles must be biocompatible and dispersible, with high stability in aqueous media under physiological conditions. In order to achieve stable colloids, many studies have reported the use of biocompatible natural and synthetic materials, such as chitosan [28], citric acid [29], poly ethylene glycol [30], dextran [31], phospholipids [32] and poloxamers (pluronic F127™, PF-127) [33], as coating agents on the surface of nanoparticles. Among them, both citric acid (CA) and PF-127 have shown high stability and strong effects on *r*_1_ and *r*_2_ relaxivities [34]. A copolymer, PF-127, is often applied as a template due to amphiphilic molecule of polyoxyethylene−polyoxypropylene−polyoxyethylene (PEO–PPO–PEO) in its structure. PF-127 is capable of forming dispersible micelles with the inner shell of hydrophobic PPO and a few PEO and the outer shell of hydrophilic PEO in contact with water [35]. Iron oxide nanoparticles with PF-127 coating exhibited a very high *r*_2_/*r*_1_ ratio due to a hydrophilic surface layer that makes it dispersible in aqueous media. Citric acid with carboxylic groups can tune the *r*_1_ relaxivity of MRI contrast agents by controlling the access of water molecules to the magnetic core [24]. For instance, recently Park et al. synthesized europium-doped iron oxide (EuIO) with three different coatings of sodium citrate (Cit), alendronate sodium trihydrate (Ale) and PMAO/PEG (PP) and showed that those coatings have significant impacts on the *r*_1_ and *r*_2_ relaxivities. They also showed there is a correlation between the hydrophilicity of those coatings and the longitudinal *r*_1_ relaxivity so that the *r*_1_ relaxivity of EuIO–Cit was 13.2-fold greater than that of EuIO–PP because of the higher hydrophilicity of EuIO–Cit [24].

In this work, we combined the magnetic properties and the *T*_2_-shortening of MZF with the *T*_1_-shortening of europium-substituted MZF. We synthesized europium-substituted EuMZF, Mn_0.6_Zn_0.4_Eu*_x_*Fe_2₋*x*_O_4_ (*x* = 0.00, 0.02, 0.04, 0.06, 0.08, 0.10 and 0.15), with two different methods, hydrothermally and through coprecipitation. To make the magnetic nanoparticles biocompatible for biomedical applications, they were coated with either citric acid or PF-127. To characterize the effects of Eu substitution and coating on the microstructure, the magnetic properties and relaxivities were investigated.

## 2. Materials and Methods

### 2.1. Materials

To synthesize the samples, the following chemicals were used: FeCl_3_·6H_2_O, MnCl_2_·4H_2_O and Zn(NO_3_)_2_, all with minimum purities of 99% (Merck Co., Rahway, NJ, USA), sodium hydroxide (NaOH) with a minimum purity of 98% (Alpha Aesar Co., Ward Hill, MA, USA); Eu(NO_3_)_3_·6H_2_O with purity of 99% (Sigma Aldrich, Burlington, MA, USA); and citric acid (C_6_H_8_O_7_) with a minimum purity of 99.5% (Unichem, Maharashtra, India).

### 2.2. Coprecipitation Synthesis of Mn_0.6_Zn_0.4_Eu_x_Fe_2−x_O_4_

Based on the proposed stoichiometry, proper amounts of the raw materials, FeCl_3_·6H_2_O, MnCl_2_·4H_2_O, Zn(NO_3_)_2,_ Eu(NO_3_)_3_·6H_2_O and citric acid, were dissolved separately in double-distilled water and stirred on a hot-plate magnetic stirrer at 80 °C until homogeneous transparent solutions of each metal salt and citric acid were obtained. Subsequently, all solutions and citric acid were mixed together and stirred for another 20 min. A boiling solution of NaOH (2M) was prepared, the mixture was poured entirely into the boiling NaOH and stirred and heated on the hot plate for 2 h more. Therefore, europium-substituted Mn–Zn ferrite phases were precipitated, based on the following reaction:0.6Mn^2+^ + 0.4Zn^2+^ + (2 − *x*)Fe^3+^ + (*x*)Eu^3+^ + 8OH^−^ → Mn_0.6_Zn_0.4_Eu*_x_*Fe_2−*x*_O_4_ + 4H_2_ O(1)

Finally, each obtained black precipitation was washed several times with double-distilled water and dried at 70 °C for 5 h. 

### 2.3. Hydrothermal Synthesis of Mn_0.6_Zn_0.4_Eu_x_Fe_2−x_O_4_

To synthesize Mn_.6_Zn_.4_Eu*_x_*Fe_2−*x*_O_4_ with different *x* values (*x* = 0.00, 0.02, 0.04, 0.06, 0.08, 0.10 and 0.15) hydrothermally, adequate amounts of Mn^2+^, Zn^2+^, Fe^3+^ and Eu^3+^ salts were completely dissolved in double-distilled water. Then, 15 mol% of citric acid to total metal salts was added to the salt solution before precipitation. After 30 min of vigorous stirring on a hot-plate magnetic stirrer at 80 °C to obtain a clear solution, a 100 mL solution of NaOH (2M) was added dropwise to the salt solution until a pH value between 10 and 12 was reached. Upon adding NaOH, the salt solution started to precipitate and some compositions of Fe^3+^, Mn^2+^, Zn^2+^ and Eu^3+^ were formed. Vigorous stirring continued for 30 min to obtain a reddish brown slurry. The solution was transferred into a Teflon-lined stainless-steel autoclave (100 mL). The autoclave was kept at 170 °C for 15 h in an electrical oven. A black precipitate was obtained and washed several times with double-distilled water to remove excess ions and finally dried at 60 °C for 10 h. 

In this method, citric acid was added to the mixture of the salt solution before the precipitation step to control metal ion coprecipitation and achieve a very homogeneous solution [36]. Citric acid can activate carboxylic groups on the surfaces of nanoparticles, making them biocompatible and dispersible in aqueous media by increasing the hydrophilicity, which is important for future application in biomedicine [24].

### 2.4. PF-127 Coating

MNPs were also coated with PF-127, as it is a copolymer, which can improve the biocompatibility of the proposed contrast agent. Briefly, a solution of PF-127 was prepared by dissolving 2 mg in 100 mL double-distilled water and heating to 50 °C on a hot-plate stirrer. To disperse dried nanoparticles, first they were treated with HNO_3_ acid under sonication. Then Fe(NO_3_)_3_ and distilled water added, and the solution was boiled, followed by continuous sonication. The dispersed nanoparticles were added to the solution containing PF-127, and vigorous stirring continued for 12 h at room temperature. The samples were centrifuged and washed several times with double-distilled water to remove excess PF-127. Figure 1 shows a schematic illustration of coating MNPs with CA and PF-127.

### 2.5. X-ray Diffraction 

An X-ray diffractometer (Bruker, D8 ADVANCED) with CuKα radiation (λ = 1.5406 Å) was used for phase identification. Measurements were carried out at diffraction angles between 2θ = 15 and 80°. To estimate mean crystallites’ sizes and microstrains, the Williamson–Hall method [37] was applied:(2)$$\beta \;=2\varepsilon \cdot \tan\theta +\frac{0.9\lambda }{D\cdot \cos\theta }\;\;y=\;\beta \cdot \cos\theta \;=2\varepsilon \cdot \sin\theta +\frac{0.9\lambda }{D}=ax+b$$
where β is the XRD peaks’ broadening (full width at half maximum, FWHM) obtained from the spinel structure diffraction peaks at 2θ, ε is the microstrain, and *D* is the mean crystallite size. Plotting *y* = βcosθ as a function of *x* = 2sinθ, mean crystallite sizes were estimated from the intercept *b* = 0.9λ/*D*, and ε from its slope after linear fitting.

### 2.6. Transmission Electron Microscopy 

The mean particle sizes of the samples were estimated from TEM micrographs, using a LEO CEM 912 transmission electron microscope with an acceleration voltage of 120 keV. A drop of suspended MNPs in water was placed on an electron-transparent carbon-coated copper grid. Then the size distribution was gained by measuring the diameter of 100 different particles. Finally, data were fitted to a log-normal distribution to calculate the mean particle size.

### 2.7. Magnetic Characterization

#### 2.7.1. Vibrating Sample Magnetometer 

A vibrating sample magnetometer (VSM, MSE-EZ9, Microsense, Lowell, MA, USA) was used to record variations of magnetization (M) with respect to the applied magnetic field (H) up to 15 kOe at room temperature. About 5 mg of dried nanoparticles were transferred to measurement vials, and the exact mass was noted, then the magnetization was computed in emu/g.

#### 2.7.2. Curie Temperature

To determine the Curie temperatures of the samples, the magnetic thermogravimetry (TG/M/) method [38] was used. In this method, magnetic force, exerted on the hanged sample in a static magnetic field produced by a hard ferrite, is measured using a Faraday balance. The temperature was increased by an electrical furnace at a rate of 5 °C/min up to above the Curie temperature and measured by a K-type thermocouple. The variation in the exerted magnetic force, which is proportional to the magnetization, with respect to temperature was recorded using Cassy Lab software.

### 2.8. Fourier-Transform Infrared Spectroscopy

FTIR spectra were recorded between 500 and 4000 cm^−1^ by a Nicolet IS 10 FT-IR Spectrometer with an attached smart iTR unit.

### 2.9. Elemental Analysis

An inductively coupled plasma optical emission mass spectrometry unit (Spectroblue Spectro/Ametek) with argon plasma was used to measure the concentrations of Fe and Eu in the final solutions. Briefly, a given volume of the suspended MNPs in double-distilled water was dissolved in nitric acid and stirred at 60 °C for 2 h. Fe and Eu standard solutions (Certipur^®^ standard solution of Fe and Eu, 1000 ppm) were used to calibrate the unit. The ion concentration was measured with the emission of Fe (λ = 238.204 nm) and Eu (λ = 381.967 nm) ions.

### 2.10. Colloidal Stability

Colloidal properties of the 0.5 mM suspended MNPs in water were characterized by Dynamic Light Scattering (DLS) using Photon Correlation Spectroscopy (PCS). The measurement is based on the fluctuations of the scattered laser light as the particles diffuse in the environment (water or other dispersion media). The measurements were carried out at different angles between 0 and 150°.

### 2.11. Relaxometery Measurements

Solutions of EuMZFs with different concentrations between 0.01 and 0.3 mM were prepared, using Phosphate Buffered Saline (PBS) solution to maintain pH near 7.4 (similar to physiological condition), and then their relaxivities *r*_1_ and *r*_2_ were measured. One milliliter of each sample was placed in a microtube, which was then placed in a container filled with water. A clinical 3T MRI system (MAGNETOM Prisma Fit, Siemens AG, Erlangen, Germany) was used to measure T_1_ and T_2_ relaxation times. All measurements were carried out with a 20-channel head coil (Siemens AG, Erlangen, Germany) at room temperature. To measure the longitudinal relaxation time *T*_1_, a segmented Turbo FLASH sequenc was used with the following parameters [39]: TR = 4.6 ms, TE = 2.0 ms, FOV = (180 mm)^2^, 180 × 180 matrix size, ${\mathrm{BW}}_{\mathrm{px}}$ = 390 Hz, flip angle = 8°. The measurements were repeated for 9 different saturation recovery times (TS): 100, 200, 300, 400, 500, 750, 1250, 2500 and 5000 ms. 

To measure the transverse relaxation time T_2_, a multi-echo spin echo pulse sequence with a Carr–Purcell–Meiboom–Gill (CPMG) [40,41] scheme was used with the following parameters: FOV = 121 × 150 mm², acquisition matrix = 208 × 256, ${\mathrm{BW}}_{\mathrm{px}}$ = 490 Hz and TR = 2000 ms. The sequence acquired 32 different spin echoes between TE_1_ = 8 ms and TE_32_ = 256 ms.

In both Turbo FLASH and CPMG image series, the mean signal intensity *S* of each sample was determined in a region of interest (ROI) that was drawn manually in the images. The longitudinal relaxation time *T*_1_ was obtained by fitting the signal to the saturation recovery equation [39]:(3)$$S\left(TS\right)=S_{0}\left(1-e^{-TS/T_{1}}\right)$$

The transverse relaxation time *T*_2_ was calculated from the CPMG series by fitting an exponential signal decay [40]: (4)$$S\left(TE\right)=S_{0}\cdot e^{-TE/T_{2}}$$

From the relaxation times, the relaxivities *r*_1_ and *r*_2_ were calculated by plotting their inverse, 1/*T*_1_ and 1/*T*_2_, as a function of the contrast-agent concentration *C*:(5)$$R_{i}=\frac{1}{T_{i}}=\frac{1}{T_{i,o}}+\frac{1}{T_{i}^{CA}}=R_{io}+r_{i}\cdot C\;\mathrm{with}\;i=1\;\mathrm{or}\;2$$
where $T_{i,o}=R_{io}{}^{-1}$ denotes the relaxation time of the solvent in the absence of the contrast agent.

## 3. Results and Discussion

### 3.1. Structural Characterization

#### 3.1.1. X-ray Diffraction Study

Figure 2a shows XRD patterns of the pure MZFs, synthesized by the coprecipitation and hydrothermal methods. As can be seen, all main peaks on both patterns are related to a cubic spinel phase (JCPDS file No. 96-200-9104) with $\mathrm{Fd}\bar{3}m$ space group. Diffraction peaks at 2θ at 30.1, 35.4, 42.8, 53.0, 56.8 and 62.2° correspond to reflection planes at (220), (311), (400), (422), (511) and (440). Peaks obtained from hydrothermal synthesis were sharper and higher in intensity than from coprecipitation. The crystallite size for the hydrothermal samples was 32 nm, while for the coprecipitation samples, it was 13.4 nm. Due to the higher crystallinity of the nanoparticles synthesized with the hydrothermal method, in the following, we investigated just these nanoparticles for further characterizations. For application in *T*2 contrast agents, nanoparticles that are synthesized at high pressure and temperature and with better crystallinity and higher core size are more favorable. It is well known that the saturation magnetization and thus the r_2_ relaxivity of magnetic nanoparticles are highly size-dependent and increase with improving crystallinity [33]. As can be seen in Figure 2b, after acid treatment and coating the nanoparticles with PF-127, an extra weak peak about 32° appeared that could be attributed to the formation of hematite because of oxidation in the presence of nitric acid. It is well known that synthesis parameters such as pH and temperature play significant roles in achieving single-phase Mn–Zn spinel ferrite nanoparticles with a high crystallinity [42,43]. The results showed that the optimum pH to obtain single-phase nanoparticles of EuMZF was between 10 and 12. At lower pHs (<10), lower crystallinity and an extra phase of hematite were observed due to incomplete precipitation (Figure 2c). In the hydrothermal process, temperature plays a significant role in the formation of spinel ferrite [42]. Our results also showed that the optimum temperature to obtain single-phase spinel ferrites of EuMZF was between 170 and 190 °C (Figure 2d). Impurities were also formed at both temperatures lower than 170 °C and higher than 190 °C.

Figure 3 shows XRD patterns of the samples with different Eu contents. As can be seen, by substituting larger Eu^3+^ ions for smaller Fe^3+^ ones, XRD patterns did not change (note the vertical lines on the Figure 3), which means their crystal structures did not change. Thus, the cubic spinel structure of EuMZF was completely unchanged, and there are mostly no traces of secondary phases. Besides, no significant shift toward lower or higher angles was observed after substituting europium into the structure of the MZF, proving that the Eu^3+^ ions entered the spinel structure in octahedral sites. Note that Eu^3+^ ions with a larger radius of about 1.07 Å show a preference to occupy the octahedral sites of Fe^3+^ with an ionic radius of 0.67 Å, forcing Fe^3+^ions to occupy tetrahedral sites. The estimated values of the crystallite size and lattice constant are listed in Table 1. Referring to Table 1, we noticed that lattice constants did not change significantly with an increase in Eu^3+^ substitution from 0 up to 0.1. The crystallite size of the pure MZF particles is 32 nm, which decreased due to the introduction of Eu into the MZF structure and reached 23 nm for *x* = 0.1. 

#### 3.1.2. TEM Analysis

Figure 4a shows the TEM images of the EuMZF NPs obtained by the hydrothermal procedure at 180 °C, pH = 12 and for *x* = 0.06. It can be observed that the particles are cubic and edged, with sizes ranging from 5 to 35 nm. Mean particle sizes calculated from TEM data were 21 ± 7 nm (Figure 4b). A comparison between these results and those obtained from X-ray analysis for the crystallite sizes, Table 1, suggests that each particle is composed of a single crystallite. The difference between the estimated values of mean particle size for sample *x* = 0.06 and that obtained from the XRD test is related to the device and to experimental errors. The Williamson–Hall method of estimating crystallite size also has some amount of errors.

#### 3.1.3. Fourier-Transform Infrared Spectra

To show attaching citric acid and PF-127 on the surface of the nanoparticles, FTIR spectra were recorded from 500 to 4000 cm^−1^. FTIR spectra of uncoated MZF, coated with citric acid and coated with citric acid plus PF-127, are illustrated in Figure 5a, and those related to Eu-substituted ones, EuMZF (*x* = 0.02, 0.06, 0.1), are shown in Figure 5b. The peaks located around 546 cm^−1^ on all samples’ FTIR spectra are attributed to the metal skeleton vibration on tetrahedral sites [44]. With substituting Eu^3+^ ions into MZF, the position of the peaks partially shifted toward higher wavenumbers due to the lattice distortion after substituting larger Eu^3+^ ions for the smaller Fe^3+^ ones. The peak around 3370 cm^−1^ is attributed to the O-H bond of water molecules absorbed by the samples [45].

After coating nanoparticles with citric acid, the peak around 1633 cm^−1^ disappeared, while a peak around 1566 cm^−1^ appeared, which is related to C=O vibration from the -COOH group of citric acid and can be assigned to the linkage of citric acid to the surface of nanoparticles [33]. Another peak around 1393 cm^−1^ also appeared after coating with citric acid, which can be connected to the asymmetric stretching of C-O from the -COOH group [46,47]. After attaching PF-127 to the surface of CA-coated nanoparticles, two extra main peaks were observe:; one around 1089 cm^−1^ related to the C-O-C bond of PEO and PPO chains of PF-127 and the other around 2865 cm^−1^, which is attributed to the C-H vibration of PF-127 [34].

### 3.2. Magnetic Properties

The room-temperature magnetic properties of the samples were investigated via a VSM unit under a maximum applied magnetic field of 15 kOe. Figure 6a shows the M-H curves of the citrate-coated samples with different Eu^3+^ substitutions. Even for high magnetic fields, the magnetizations did not saturate. Therefore, for the high field parts of the M-H curves, the variations of M versus 1/H were plotted, fitted linearly and extrapolated for 1/H→0 to determine the saturation magnetizations. The corresponding values of saturation magnetizations (*M_s_*) are listed in Table 1. The low *H_c_* values of the samples with the higher Eu^3+^ substitutions, which are in the range of measurement errors (0.4 and 0.12 Oe for *x* = 0.10 and 0.15, respectively), show that they are nearly superparamagnetic. Bulk magnetic materials comprise magnetic multidomains, with a preferred direction in each magnetic domain, that are separated by a wall, named the Bloch wall or domain wall. Below a critical size (*r_c_*), the formation of domain walls needs more energy than that of a single domain. At these sizes, thermal effects are enough to spontaneously demagnetize a previously saturated assembly of magnetic nanoparticles. This behavior is named superparamagnetism [48]. There is a decreasing trend in the obtained values of *H_c_* with an increase in Eu substitution, i.e., *x*, which can be attributed to a reduction in magnetocrystalline anisotropy. The *H_c_* for *x* = 0.00 was 12Oe, which decreased to 0.12Oe for *x* = 0.15. According to the Stoner–Wohlfarth theory, coercivity is related to the anisotropy constant (*K*) based on the following equation [49]:(6)$$K=\frac{M_{s}\times H_{c}}{0.98}$$

Coercivity shows the strength of the magnetic field to overcome the anisotropy barriers and displace the domain wall, in order to align the magnetization of the nanoparticles with the magnetic field direction [50]. By increasing *x* from 0.00 to 0.15, the anisotropy constant decreased from 673 to 6 ergs*/*cm^3^, which can be related to distorted crystal ferrite structure due to substituting larger Eu^3+^ ions for Fe^3+^ ions in B sites. In Neel’s model [51], three interactions between magnetic moments in tetrahedral (A) and/or octahedral (B) sites, A–A, B–B and A–B, in spinel ferrites are proposed. Among these interactions, A–B is much stronger than the others. It is well known that Fe^3+^ ions can occupy both octahedral and tetrahedral sites, and the magnetic properties of some ferrites like MZF are governed by the A–B exchange interaction of iron ions [52]. Thus, due to the lower magnetic moment of Eu^3+^ (3µB) than Fe^3+^ ions (5µB), the replacement of Fe^3+^ ions by Eu^3+^ ions interrupts the A–B exchange interactions, leading to reduced values of *H_c_* and *M_r_* of EuMZF. Similar results for coercivity and magnetization were previously obtained for Eu^3+^-doped cobalt ferrite [53] and cobalt-doped iron oxides [54]. As can be seen from Figure 6a and Table 1, an increase in Eu substitution leads to a decrease in *M_s_*, except for *x* = 0.02, where *M_s_* has increased compared to *x* = 0. An increase in *M_s_* at very low *x* has also been reported for other rare-earth-substituted spinel ferrites in the literature [55]. For pure MZF, *M_s_* is 55 emu/g, while for *x* = 0.15, it is 45 emu/g. In two sublattice ferrimagnetic materials, the ions can occupy two available sublattices, tetrahedral A and octahedral B sites. The magnetic moments *M* of these two sites are aligned anti-parallel with each other, and then the net absolute value of the magnetization is:(7)$$M=\left|M_{B}-M_{A}\right|$$

Thus, the net magnetization of EuMZF is governed by the cation distribution of Fe^3+^, Mn^2+^ and Zn^2+^ ions in A and B sites. It was shown that rare-earth ions prefer to occupy the octahedral B sites due to their larger ionic radii [56]. As Eu^3+^ ions enter the B sites, some Fe^3+^ ions are replaced and forced to migrate to the A sites. That increases A-site magnetization $\;M_{A}$ and consequently decreases the net magnetization *M*, Equation (7). M–H curves of the samples with different Eu substitutions and coated with both citric acid and PF-127 are shown in Figure 6b. The addition of the second PF-127 coating led to a slight reduction of the saturation magnetization *M_s_*, which can be attributed to the presence of the non-magnetic PF-127 layer on the surface of nanoparticles that reduces the magnetic inter-particle interactions [57,58]. 

Figure 7 illustrates the variation of magnetization versus temperature (M–T) for different Eu-substituted samples. For all samples, as the temperature increases, the magnetization decreases monotonically and approaches zero at the Curie temperature, whereby the samples lose their magnetic ordering and show paramagnetic behavior. The Curie temperatures were estimated by extrapolating the linear part of M–T curves to the temperature axis and summarized in Table 1. There is a clear hump on all curves especially on the pure MZF (*x* = 0) curve. One possible explanation for the observed hump on the M–T curves could be the simultaneous presence of two or more spinel phases with similar XRD patterns in the structure of EuMZF, which can be then integrated into a single phase after heating the samples up to temperatures above 180 degrees. By increasing Eu substitutions, a decrease in Curie temperature is observed, which can be explained by the weakening of exchange interactions [59]. The magnetic characteristics of MZFs are controlled by the Fe–Fe (3d–3d) exchange interaction [60]. As discussed before, an increase in the concentration of the Eu^3+^ reduces net magnetization in the B sublattice. Due to the weakening of exchange interaction, less thermal energy is needed to randomize magnetic moments, so the Curie temperatures decrease [59]. 

### 3.3. Colloidal Stability

To investigate the colloidal stability of nanoparticles, an important factor to determine the biocompatibility of nanoparticles for their future applications in biomedicine, the hydrodynamic sizes of the samples, were determined DLS Method. As shown in Figure 8, the size distribution at different angles for citric-acid-coated samples was relatively narrow; the average hydrodynamic sizes of the samples coated with citric acid (*x* = 0, *x* = 0.15) and PF-127 were 80 ± 18 (PDI = 0.23), 93 ± 30 (PDI = 0.33) and 195 ± 65 nm (PDI = 0.33), respectively. Often hydrodynamic measurements are only done at 90° to minimize backscattering due to instrument setup and other errors. The hydrodynamic diameter of the three samples at 90° are 71.7 ± 0.7 (PDI = 0.01), 79.8 ± 2.5 (PDI = 0.031) and 149.4 ± 2.6 nm (PDI = 0.017). The hydrodynamic size of the sample coated with PF-127 is about two times larger than that of the samples coated with citric acid. The higher hydrodynamic sizes of the samples with pluronic coatings are due to their higher tendencies for aggregation than those coated just with citric acid. At a low concentration of pluronic and low temperature, a copolymer of pluronic is formed as single unimers. Stable micelles are formed when the concentration increases and reaches critical micelle concentration (CMC). As the micelles exceed its CMC, nanoparticles tend to aggregate, which leads to larger hydrodynamic sizes [33]. In the case of citric-acid-coated samples, the sizes were comparable to mean particle size and crystallite size, which indicates that magnetic nanoparticles are not so agglomerated.

### 3.4. Relaxometery Measurements 

#### 3.4.1. Effect of Surface Design on Relaxivities

To increase *r*_1_ and *r*_2_, designing the nanoparticles’ surfaces and consequently, controlling the water accessibility of the magnetic core is a crucial factor. Figure 9a,b illustrate the variation of the relaxation rates *R*_1_ and *R*_2_ as a function of [Fe+Eu] concentration for citric-acid- and PF-127-coated Mn_0.6_Zn_0.4_Eu_0.15_Fe_1.85_O_4_. As can be seen, there are explicitly linear relations between *R*_1_ and *R*_2_ and concentration for all samples, so the slopes represent the specific *r*_1_ and r_2_ relaxivities. The citric acid coating resulted in a relatively high *r*_1_ (note the slope of the *R*_1_–concentration fitting curve for the citric-acid-coated sample in Figure 9a), which can be explained by the high hydrophilicity of the small citric acid molecules that increases the accessibility of Eu ions in the EuMZF structure to the surrounding water molecules. By applying the second coating layer of PF-127, *r*_1_ decreased from 11.60 to 3.35 mM^−1^s^−1^ as a result of the larger hydrodynamic diameter of the PF-127-coated EuMZF rather than the citric-acid-coated one, leading to the reduced accessibility of Eu ions to the hydrogen pool [24]. The low *r*_1_ of the PF-127-coated samples can be also related to the lower hydrophilicity and the long chain of PEO–PPO–PEO in PF-127 that limits the permeability of the magnetic core to water molecules. The hydrophilicity is directly related to *r*_1_ as it controls the water coordination number and the accessibility of the magnetic core to the water pool. The *r*_2_ value for citrate alone and citrate plus PF-127-coated Mn_0.6_Zn_0.4_Eu_0.15_Fe_1.85_O_4_ were 130 and 152 mM^−1^s^−1^, respectively. For nanoparticles with both coatings, *r*_2_ is slightly larger than that for nanoparticles with just a citric acid coating, which might be related to the contribution of PF-127 to *T*_2_ relaxation [34].

#### 3.4.2. Effect of Eu Substitution on Relaxivities

Figure 10a shows *T*_1_-weighted MR-images of the EuMZF nanoparticles coated with citric acid and different Eu substitutions (*x* = 0.00, 0.04, 0.08, 0.15). As can be seen, pure MZF (*x* = 0.00) does not increase the signal intensity of *T*_1_-weighted MR-images with increasing [Fe+Eu] concentrations (positive contrast). However, as *x* increases from 0.00 to 0.15, the samples show a reduction in *T*_1_, resulting in higher signal intensities. So for *x* = 0.15, we can see a significant positive contrast enhancement in the *T*_1_-weighted MRI, suggesting that Mn_0.6_Zn_0.4_Eu_0.15_Fe_1.85_O_4_ is a potential candidate as a positive MRI contrast agent. The *r*_1_ relaxivities were calculated by plotting the inverse *T*_1_ as a function of [Fe+Eu] concentration (Figure 10c). In Table 2, we can see that *r*_1_ values grow with an increase in *x*: r_1_ for pure MZF is about 1.73 mM^−1^s^−1^, while for *x* = 0.15, that is about 11.6 mM^−1^s^−1^. This increase in *r*_1_ can be attributed to the shielding effect of paramagnetic Eu^3+^ ions with six unpaired electrons against the main magnetic field [61]. The shielding effect of paramagnetic ions around the water protons decreases the magnetic field sensed by protons and, consequently, causes water protons to resonate at lower frequencies (chemical shift effect). With decreasing Larmor frequency, the frequencies of the water protons associated with the molecular motion in the surrounding medium can match better, leading to an efficient transfer of energy and shorter *T*_1_ (larger *r*_1_). In EuMZF, the dipole–dipole interaction between the paramagnetic centers of Mn^2+^ and Eu^3+^ ions and water protons in inner- and outer-sphere shells is responsible for the high *r*_1_ relaxivity. There are two main relaxivity mechanisms: inner- and outer-sphere relaxations. In the inner-sphere relaxation mechanism, water protons are directly bound to metal ions, and energy is released during the relaxation process to the water pool through water molecular exchange. In the outer-sphere relaxation mechanism, water protons close to paramagnetic centers can also relax. High-spin metal ions like Gd^3+^ with *S* = 7/2 and Eu^3+^ with *S* = 6/2 can accelerate water-proton relaxation rates through dipole–dipole interactions [62]. The unpaired electrons of a paramagnetic center around the protons can create a local magnetic field, which can oppose the main magnetic field and shield protons against the main applied magnetic field (B_0_). The dipole–dipole interactions between protons spins and unpaired electrons of paramagnetic Eu^3+^ ions govern the longitudinal relaxation rate of the water molecules.

Figure 10b shows *T*_2_-weighted images of EuMZF with different *x* values and [Fe+Eu] concentrations. All samples show an excellent negative contrast that increases with [Fe+Eu] concentrations, indicating that pure MZF and Eu-substituted MZF, as negative contrast agents, can effectively decrease the *T*_2_ of water protons. The measured *T*_2_ values were inversely plotted as a function of [Fe+Eu] concentration (Figure 10d). Then *r*_2_ values were extracted from the slope of the fitted lines and listed in Table 2. According to Table 2, the measured *r*_2_ first increased from 263 mM^−1^s^−1^ for pure MZF to 300 mM^−1^s^−1^ for *x* = 0.02, then decreased to 130 mM^−1^s^−1^ for *x* = 0.15. The increase in *r*_2_ for low *x* values and the decrease for higher *x* ones are comparable to changes in saturation magnetization, as *r*_2_ is proportional to the *M_s_²* directly. Some magnetic and relaxivity properties of MZF nanoparticles with different coatings are listed in Table 3. The *r*_1_ relaxivities of EuMZF are relatively higher than those reported by others, but the *r*_2_ one is in the same range. The *r*_2_/*r*_1_ ratio is also an important parameter to determine the efficiency of an MRI contrast agent. Negative contrast agents mainly possess a high *r*_2_/*r*_1_ ratio, whereas a low *r*_2_/*r*_1_ value close to one refers to a positive contrast agent. For a dual-mode MRI contrast agent, high values of *r*_1_ and *r*_2_, but a moderate *r*_2_/*r*_1_ ratio is needed [63]. Table 4 shows the main MRI characteristics of some commercial contrast agents. Herein, the *r*_2_/*r*_1_ ratio for pure MZF is 152, which is much greater than those of commercial negative MRI contrast agents [64]. This indicates that pure MZF is an excellent negative contrast agent. The superiority of the pure MZF, as a negative MRI contrast agent, over the commercial negative contrast agents can be due to the higher crystallinity of MZF nanoparticles because of synthesis at relatively high temperature and high pressure in the hydrothermal route. With increasing europium content, *r*_1_ increases, whereas the *r*_2_ and *r*_2_/*r*_1_ ratio decrease, so the *r*_2_/*r*_1_ ratio is 11.2 for *x* = 0.15. The Mn_0.6_Zn_0.4_Eu_0.15_Fe_1.85_O_4_ sample as a positive MRI contrast agent is comparable to the commercial positive MRI contrast agents, but it possesses a higher *r*_2_/*r*_1_ ratio due to a higher *r*_2_. These results indicate that the Mn_0.6_Zn_0.4_Eu_0.15_Fe_1.85_O_4_ sample with high both r_1_ and *r*_2_ and a moderate *r*_2_/*r*_1_ ratio of 11.2 is potentially a suitable dual-mode MRI contrast agent.

## 4. Conclusions

In this work, europium-substituted Mn-Zn ferrite (Mn_0.6_Zn_0.4_Eu*_x_*Fe_2−*x*_O_4_, *x* = 0.00, 0.02, 0.04, 0.06, 0.08, 0.10 and 0.15) nanoparticles were synthesized by the coprecipitation and hydrothermal methods. The hydrothermally prepared nanoparticles were single-phase spinel and have high crystallinity and good colloidal stability. The substitution of Eu did not significantly change XRD patterns and crystal structure of MZF. An increase in saturation magnetization for a low amount of Eu was observed, followed by a reduction for higher substitutions. The *r*_1_ relaxivity of EuMZF increased with increasing europium substitution. On the contrary, the transverse relaxivity decreased as the Eu substitution increased, except for low amounts of substitution. The *r*_2_/*r*_1_ ratio was very high for pure MZF but decreased with Eu substitution. The Mn_0.6_Zn_0.4_Eu_0.15_Fe_1.85_O_4_ sample had high r_1_ and r_2_ and a moderate *r*_2_/*r*_1_ ratio, indicating that pure MZF is a good candidate for negative contrast agents, while Mn_0.6_Zn_0.4_Eu_0.15_Fe_1.85_O_4_ is suitable for application as a dual-mode contrast agent. The results also show that nanoparticles coated with citric acid had higher *r*_1_ and lower *r*_2_ than those of nanoparticles coated with PF-127 correspondingly.

## Figures and Tables

**Figure 1 nanomaterials-13-00331-f001:**
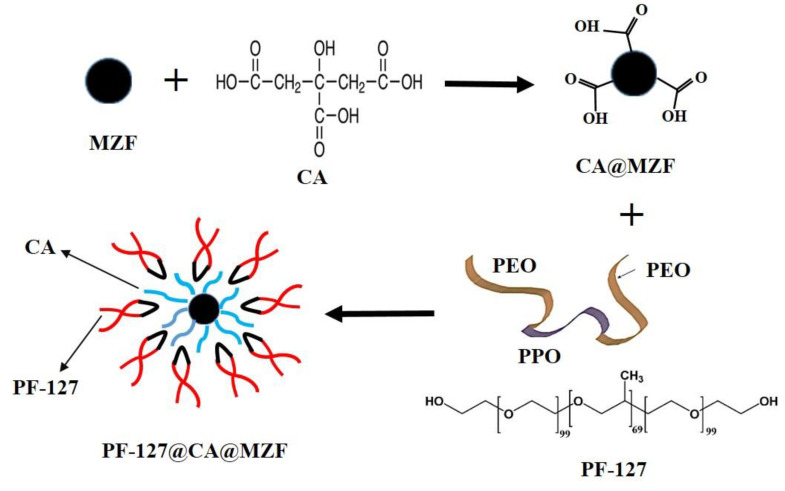
Schematic illustration for the formation of citric acid (CA) and pluronic F127 (PF-127) on the surface of MZF nanoparticles.

**Figure 2 nanomaterials-13-00331-f002:**
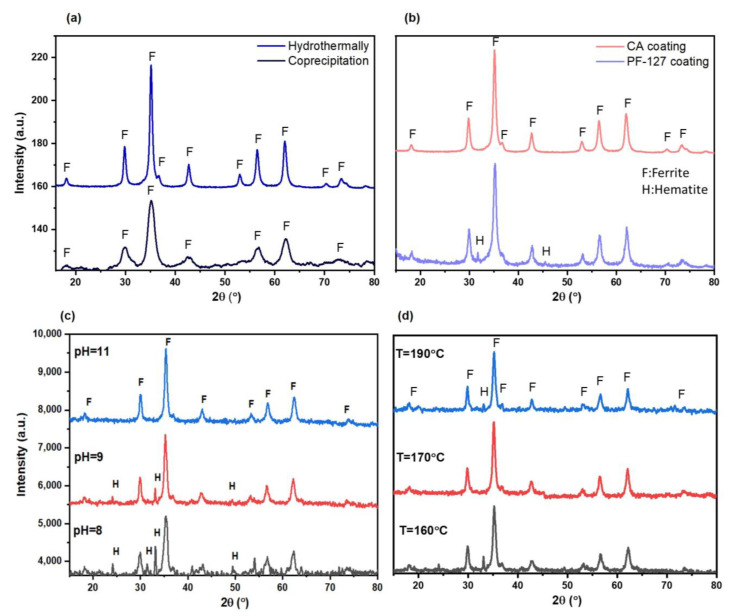
X-ray diffraction patterns of pure MZF nanoparticles synthesized (**a**) with co-precipitation and hydrothermal methods (**b**) with different coatings of citric acid (CA) and pluronic F-127 (PF-127) (**c**) at different precipitation pHs and (**d**) at different hydrothermal temperatures.

**Figure 3 nanomaterials-13-00331-f003:**
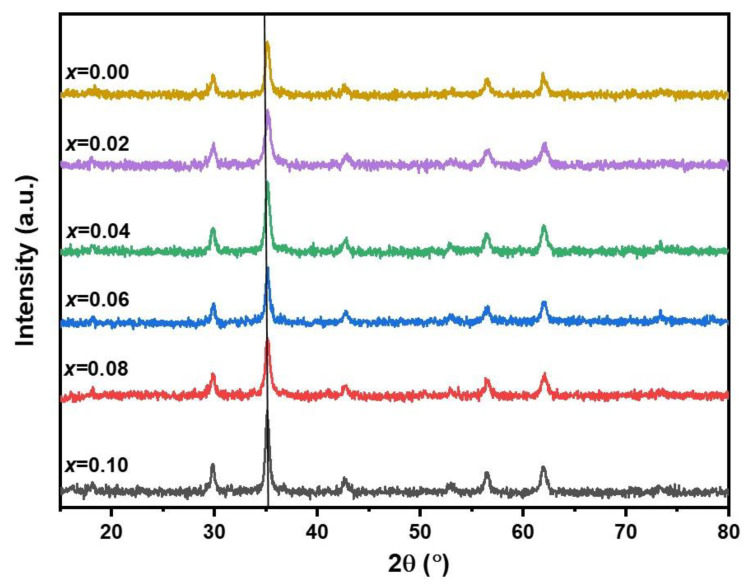
X-ray diffraction patterns of Mn_0.6_Zn_0.4_Eu*_x_*Fe_2 −*x*_O_4_ (*x* = 0.00, 0.02, 0.04, 0.06, 0.08, 0.1).

**Figure 4 nanomaterials-13-00331-f004:**
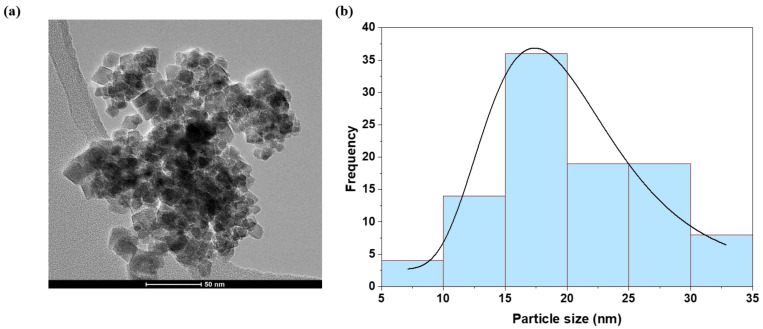
(**a**) TEM image of the sample *x* = 0.06 and (**b**) corresponding histogram of the particle size distribution.

**Figure 5 nanomaterials-13-00331-f005:**
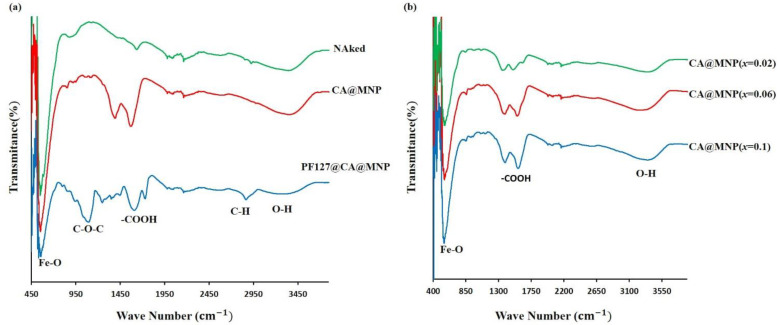
Fourier-transform infrared spectra of (**a**) naked, citric acid and pluronic F-127 coated MZF and (**b**) Eu-substituted MZF (*x* = 0.02, 0.06, 0.10).

**Figure 6 nanomaterials-13-00331-f006:**
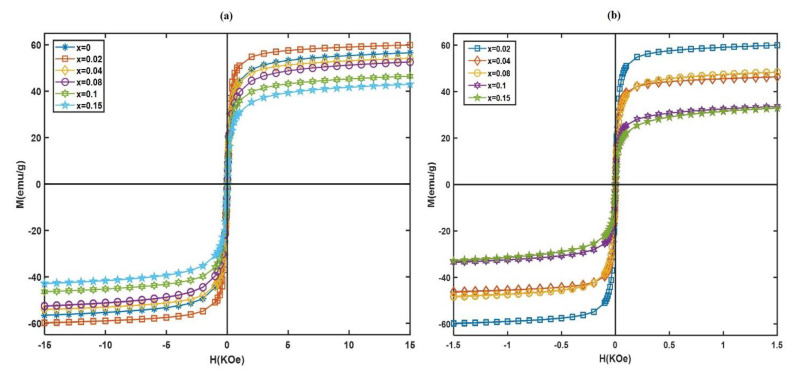
M-H curves of dry powders for Mn_0.6_Zn_0.4_Eu*_x_*Fe_2−*x*_O_4_ (*x* = 0.00, 0.02, 0.04, 0.08, 0.10, 0.15) with (**a**) citric acid and (**b**) pluronic F-127 coating.

**Figure 7 nanomaterials-13-00331-f007:**
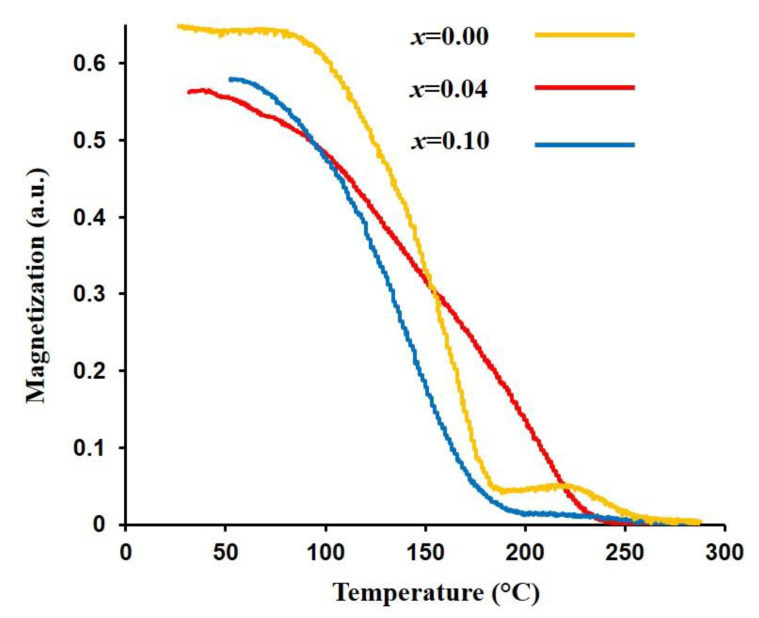
M-T curves for Mn_0.6_Zn_0.4_Eu*_x_*Fe_2−*x*_O_4_ (*x* = 0, 0.04, and 0.1).

**Figure 8 nanomaterials-13-00331-f008:**
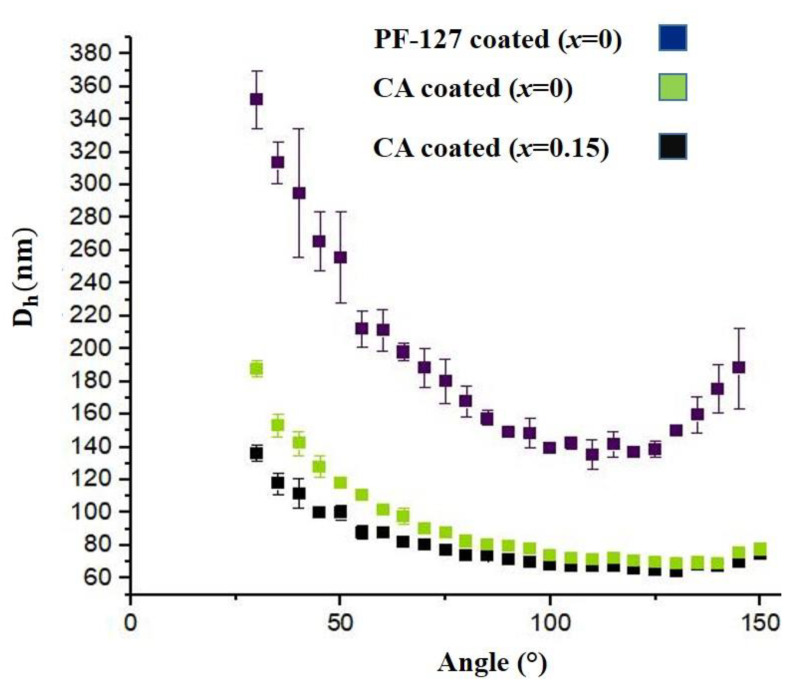
Average hydrodynamic diameter of nanoparticles dispersed in water for citric-acid- and pluronic F-127-coated EuMZF.

**Figure 9 nanomaterials-13-00331-f009:**
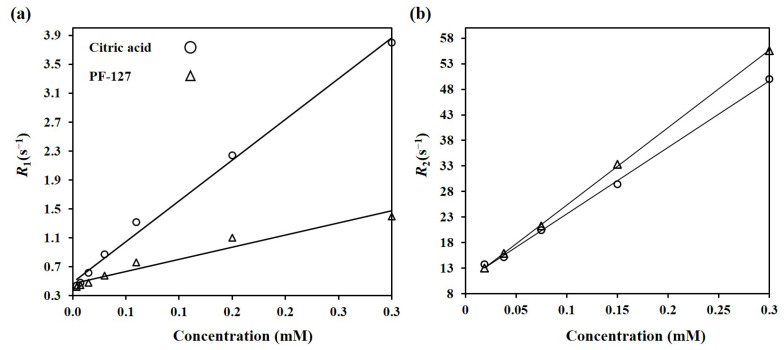
The variation in the relaxation rates (**a**) *R*_1_ and (**b**) *R*_2_ as a function of concentration for citric-acid- and pluronic F-127-coated Mn_0.6_Zn_0.4_Eu_0.15_Fe_1.85_O_4_.

**Figure 10 nanomaterials-13-00331-f010:**
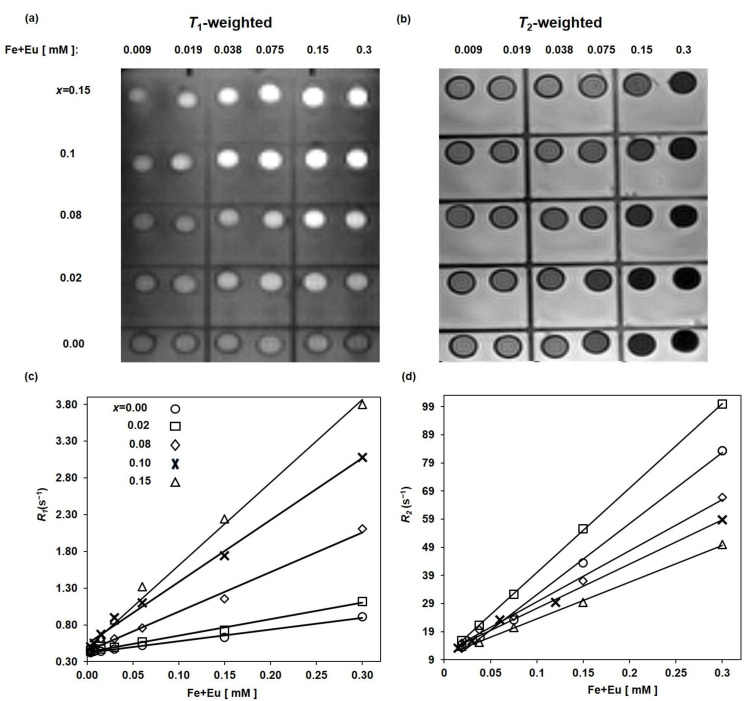
(**a**) *T*_1_-weighted MR images, (**b**) *T*_2_-weighted MR images, (**c**) longitudinal *R*_1_ relaxation rate and (**d**) transverse *R*_2_ relaxation rate as a function of concentration for Mn_0.6_Zn_0.4_Eu_x_Fe_2−*x*_O_4_ (*x* = 0, 0.02, 0.08, 0.1, 0.15).

**Table 1 nanomaterials-13-00331-t001:** Crystallite sizes, lattice constants, saturation magnetizations (*M_s_*) and Curie temperatures (T_c_) of the EuMZF nanoparticles with citric acid coating and with different values of substitutions (*x* = 0.00, 0.02, 0.04, 0.06, 0.08, 0.1 and 0.15).

*x* Values	Crystallite Size ±1 (nm)	Lattice Constant ±0.001 (Å)	$M_{s}\pm 1$ (emu/g)	T_c_ (°C)
0.00	32	8.462	55	261
0.02	33	8.466	58	-
0.04	30	8.462	56	238
0.06	28	8.466	-	-
0.08	36	8.466	54	-
0.10	23	8.457	48	183
0.15	-	-	45	-

**Table 2 nanomaterials-13-00331-t002:** Longitudinal *r*_1_ and transverse *r*_2_ relaxivities of EuMZF nanoparticles with different Eu substitutions.

*x* Values	$r_{1}$ $\left(mM^{-1}s^{-1}\right)$	$r_{2}(mM^{-1}s^{-1})$	$r_{2}/r_{1}$
0.00	1.73 ± 0.16	263 ± 31	152
0.02	2.05 ± 0.24	300 ± 70	146
0.08	5.9 ± 0.71	207 ± 19	35
0.10	8.4 ± 1.2	157 ± 16	18.7
0.15	11.6 ± 1.44	130 ± 16	11.2

**Table 3 nanomaterials-13-00331-t003:** Some magnetic and relaxivity properties of MZF with different coatings.

Reference	Structure	*M_s_* (emu/g)	$r_{1}$ $\left(mM^{-1}s^{-1}\right)$	$r_{2}$ $\left(mM^{-1}s^{-1}\right)$
Zahraee et al. [65]	MZF@Citric acid	53	4	132
Zahraee et al. [65]	MZF-PEG	49	3	79
Junzhao et al. [66]	MZF@4sPCL-*b*-P(MEO_2_MA-*co*-OEGMA)	59	1.28	138
Tayebe et al. [67]	MZF@PEG	-	-	88
Soraya et al. [30]	MZF@PEG	60	-	314

**Table 4 nanomaterials-13-00331-t004:** Main characteristics of some commercial MRI contrast agents [64,65].

Name	Structure	D_h_(nm)	B_0_(T)	$r_{1}(mM^{-1}s^{-1})$	$r_{2}(mM^{-1}s^{-1}{)}_{\;}$	*r_2_*/*r_1_*
Resovist	Carboxydextran-coated USPIOs	60	3	4.6	143	31.08
Feridex	Dextran-coated SPIOs	150	3	4.1	93	22.68
Gadomer	Gd-DTOA	30	3	13	23	1.76
Magnevist	Gd-DTPA	-	3	3.1	3.7	1.19
Ferucarbotran^®^ (SHU-555A)	Carboxydextran-coated USPIOs	60	3	7.3	57	7.8

## Data Availability

Data can be available upon request from the corresponding author.

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
