# Peer review of "Effect of Europium Substitution on the Structural, Magnetic and Relaxivity Properties of Mn-Zn Ferrite Nanoparticles: A Dual-Mode MRI Contrast-Agent Candidate"

_nanomaterials, 2023, doi:10.3390/nano13020331_

Round 1

Reviewer 1 Report

The manuscript reports the synthesis and extensive characterization of Mn-Zn ferrite nanoparticles, and how their magnetic properties change when various quantities of europium are added to the particles during their synthesis. The authors have synthesized particles with various percentages of Europium, and investigated the effect on their size, size distribution, crystallinity, surface composition, and on their magnetic properties, especially with respect to their relaxivity for NMR contrast imaging.

The manuscript is well written, and potentially interesting to a broad variety of scientists. However, some interpretation of the date need to be reconsidered, since they seem either erroneous or at least too superficial.

1)      First, the authors have synthesized both nanoparticles via coprecipitation and via hydrothermal process, but decided to focus all their characterization efforts only on the latter, because of their better crystallinity. However, this seems to be either a waste of effort, since coprecipitation is a much cheaper synthesis method, typically leading to higher yields and much more easy to upscale. Furthermore, the higher crystallinity is probably a result of the larger particles size obtained by hydrothermal synthesis. It is a pity not to have done a magnetic characterization of the particles obtained by coprecipitation too.

2)      The difference between the size estimated from TEM image analysis and the size obtained from XRD is quite large: some more explanation about this difference is necessary. Furthermore, some other method, such as SAXS, could be used to obtain a more reliable size estimate of the crystallites.

3)      The comment made about the appearance of some additional spectral line in the XRD data after functionalization with pluronics is not credible. How the attachment of a block copolymer, performed at a temperature of about 50 °C, can change the crystallinity of particles prepared at a temperature of more than 150°C?

4)      The interpretation of the colloidal stability data (DLS size) is not at all accurate. A size of abut 70 nm or more is an indication of the presence of aggregates of single crystals. And the effect of pluronics on the size of cluster is an increase of a few nm, not a doubling of their size.

Author Response

Response to Reviewers (nanomaterials-2133186)

First of all, we would like to thank the respected reviewers for their thorough and encouraging evaluation of our manuscript. In the following, we would like to address the individual points of critique together with a list of the changes made. In the annotated version of the manuscript, we have marked relevant parts that are in particular changed as a consequence of the questions and remarks of the reviewers.

Reviewer 1

R1.1: First, the authors have synthesized both nanoparticles via coprecipitation and via hydrothermal process, but decided to focus all their characterization efforts only on the latter, because of their better crystallinity. However, this seems to be either a waste of effort, since coprecipitation is a much cheaper synthesis method, typically leading to higher yields and much more easy to upscale. Furthermore, the higher crystallinity is probably a result of the larger particles size obtained by hydrothermal synthesis. It is a pity not to have done a magnetic characterization of the particles obtained by coprecipitation too.

Response: we synthesized the samples to be used as MRI contrast agents. For application in T2 contrast agents, nanoparticles which are synthesized at high pressure and temperature, thus with better crystallinity and higher core size are more favorable (Theranostics. 2018; 8(9): 2521–2548, Nanotechnology 27 (2016) 255702 (12pp)). The saturation magnetization and thus r2 relaxivity of magnetic nanoparticles are highly size-dependent and increase with improving crystallinity (Adv.Mater.2021, 33, 1906539). When the size of nanoparticles decreases and reaches below 10nm, their r2 decrease dramatically and the nanoparticles becomeT1 agent which is a different project. MZF is a T2 contrast agent and we improved its T1-shortening effect by substituting Europium.  The goal of this work is to design a dual-mode contrast agent with both high r1 and r2 relaxivities. In our next paper, we will investigate the magnetic properties and mri relaxivities of EuMZF nanoparticles synthesized with the coprecipitation method and we will compare those results with the results of hydrothermally synthesized nanoparticles.

R1.2: The difference between the size estimated from TEM image analysis and the size obtained from XRD is quite large: some more explanation about this difference is necessary. Furthermore, some other method, such as SAXS, could be used to obtain a more reliable size estimate of the crystallites.

Response: the particle size (21±7) estimated from TEM Method was for sample x=0.06. The corresponding crystallite size for this sample is 28±1. The difference between the estimated value of mean particles size for sample x = 0.06 and that obtained from XRD is related to device, and operator’s errors. The Williamson-hall method of estimating crystallite size also has some degree of errors.

R1.3: The comment made about the appearance of some additional spectral line in the XRD data after functionalization with pluronics is not credible. How the attachment of a block copolymer, performed at a temperature of about 50 °C, can change the crystallinity of particles prepared at a temperature of more than 150°C?

Response: to disperse dried nanoparticles before functionalization with pluronic, they were subjected to an acid treatment. Dried nanoparticles were treated with HNO3 under sonication. Then Fe(NO3)3 and distilled water were added and the solution was boiled followed by continuous sonication. The appearance of the extra peak in the XRD pattern could be happened during the acid treatment due to oxidation in presence of nitric acid.

R1.4: The interpretation of the colloidal stability data (DLS size) is not at all accurate. A size of about 70 nm or more is an indication of the presence of aggregates of single crystals. And the effect of pluronics on the size of cluster is an increase of a few nm, not a doubling of their size.

Response: the related discussion was modified. The higher hydrodynamic sizes of the samples with pluronic coatings are due to their higher tendencies for aggregation than those coated just with citric acid

Author Response

Response to Reviewers (nanomaterials-2133186)

First of all, we would like to thank the respected reviewers for their thorough and encouraging evaluation of our manuscript. In the following, we would like to address the individual points of critique together with a list of the changes made. In the annotated version of the manuscript, we have marked relevant parts that are in particular changed as a consequence of the questions and remarks of the reviewers.

Reviewer 2

R2.1: Page 4 line 149; the wavelength should be written: λ = 1.5406 Å. (in my printout @ is used instead of λ)

Response: The symbol ‘’λ’’ was substituted for ‘’@’’ in the appropriate place.

R2.2: Page 8 lines 274-276: I do not see a good reason for a direct relation between crystallite size and difference in atomic radii – this sentence/explanation could be removed.

Response: The sentence ‘’ the decreased values of crystallite size upon substituting Eu is attributed to the difference in atomic radii of Eu3+ and Fe3+ ’’ was removed

R2.3: Page 9 figure 4 a): The images are shown with almost the same magnification as the length bars indicate, why are both images shown? They are both taken on the same sample.

Response: The image with a scale bar of 20 nm was removed

R2.4: Magnetic properties Pages 10 – 12: Figure 6 a) the magnetic room temperature data in Table 1 are derived from these curves on the citric acid samples (this should be pointed out in the caption of Table 1). The magnetization of Ferrites (ferrimagnets) do not saturate but show a linear increase at higher fields due to alignment of the antiparallel sublattice. A better extrapolation of the excess moment |MB-MA |is obtained from linear extrapolation of the high field behavior to zero field.

Response: the caption of table one was updated. The magnetization of nanomaterials doesn’t saturate but the magnetization of bulk ferrite saturates. We plotted high filed part as a function of 1/H and extrapolated when 1/H®0. So magnetization can be acquired when the magnetic field tends to be infinite, which can show saturation magnetization. This method was applied for Nano ferrite in our previous publication (Scientific Reports (2021) 11:16795) and other references (Journal of Solid State Chemistry 213 (2014) 57–64)

R2.5: Figure 6 a) Coercivity assuming that the particles are in superparamagnetic states at room temperature, the coercivity should be zero, which agrees quite well with the very small (negligible) values quoted in table 1. I think that the Hc, Mr and K values could be removed from Table 1.

Response: the values of Hc, Mr and K were removed from table 1 and we mentioned some of them in the magnetic properties section as we discussed them before.

R2.6: Figure 7 and the derived values of Tc quoted in Table 1: the value that can be extracted from the x=0.04 curve yields Tc~240 , not 190 K as quoted in Table 1. (The value of Tc that can be derived for x=0.10: ~180 K is not quoted in Table 1. There is no consistent decreasing trend of Tc with x, thus the text on this should be modified.

Response: the values of Tc were modified. Some of the values were not written in the appropriate position. For x=0.04, Curie temperature was derived again.  For x=0 curve, the value of Tc was determined again from the second linear part of the curve. For x=0.1, Tc was estimated from the corresponding curve. For all of the other EuMZF samples, the values of Tc were not estimated.  So the modified values of Tc for x = 0.00, 0.04, and 0.10 are 261, 238, and 182 °C respectively.  After modifying the values of Tc, the decreasing trend is still observed.

R2.7: Page 11 lines 374-376: it is not likely that a redistribution of cations in A and B sites occur at temperatures below 300 C.

Response: the corresponding discussion was removed and the following discussion was added ‘One possible explain for the observed hump on the M-T curves could be the simultaneous presence of two or more spinel phases with the similar XRD patterns in the structure of EuMZF which can be then integrated into a single phase after heating the samples up to the temperatures above 180 degrees.’’

Reviewer 3 Report

The authors have presented a detailed description of the synthesis and the characterizations of europium-substituted Mn-17 Zn ferrite (Mn0.6Zn0.4EuxFe2-xO4, x = 0.00, 0.02, 0.04, 0.06, 0.08, 0.10, and 0.15, which here named MZF for x = 0.00 and EuMZF for others). The current results indicated that the samples with citric acid coating had higher r1 and lower r2 relaxivities than those of Pluronic F127 coated samples. This work is nice, but the below points should be addressed.

1.     Please provide the XPS and MAPPING

2.     Please provide the PDI for the DLS data.

3.     The authors have stated, “In general, two different types of MRI contrast agents exist. Positive ones (e.g. based on Gadolinium chelates), which predominantly shorten the longitudinal relaxation time T1, so that the signal intensity in the region of interest is increased (hyperintensity in T1- weighted images).” This part should be highlighted and cited related refs, such as Colloid Surface B, 2022, 213, 112432; Dalton Transactions, 2022, 51, 14817-14832. In the IR discussion, “Another peak around 1393 cm-1 appeared too after coating with citric 304 acid, which can be connected to asymmetric stretching of C-O from the -COOH group., some refs could be added, such as Inorganics, 10(2022) 202 and Micropor. Mesopor. Mat, 341(2022) 112098.

4.     I HAVE found some documents on the materials of europium-substituted Mn-Zn ferrite, please list a Table that will highlighted your work, which will be interested in the reader.

Author Response

Response to Reviewers (nanomaterials-2133186)

First of all, we would like to thank the respected reviewers for their thorough and encouraging evaluation of our manuscript. In the following, we would like to address the individual points of critique together with a list of the changes made. In the annotated version of the manuscript, we have marked relevant parts that are in particular changed as a consequence of the questions and remarks of the reviewers.

Reviewer 3

R3.1: Please provide the XPS and MAPPING

Response: Currently we don’t have the result of ‘’ XPS and MAPPING’’ and as you know right now because of the new year holiday it is not possible to get the result very soon. We will present the results of XPS and MAPPING in our next paper.

R3.2: Please provide the PDI for the DLS data.

Response: the PDI for the DLS data was added in the appropriate place.

R3.3: The authors have stated, “In general, two different types of MRI contrast agents exist. Positive ones (e.g. based on Gadolinium chelates), which predominantly shorten the longitudinal relaxation time T1, so that the signal intensity in the region of interest is increased (hyperintensity in T1- weighted images).” This part should be highlighted and cited related refs, such as Colloid Surface B, 2022, 213, 112432; Dalton Transactions, 2022, 51, 14817-14832. In the IR discussion, “Another peak around 1393 cm-1 appeared too after coating with citric 304 acid, which can be connected to asymmetric stretching of C-O from the -COOH group., some refs could be added, such as Inorganics, 10(2022) 202 and Micropor. Mesopor. Mat, 341(2022) 112098.

Response: the suggested references of reviewers were inserted into the mentioned parts and the corresponding parts were highlighted.

R3.4: I HAVE found some documents on the materials of europium-substituted Mn-Zn ferrite, please list a Table that will highlighted your work, which will be interested in the reader.

Response: we checked other papers with the title of ‘’ europium-substituted Mn-Zn ferrite’’. Unfortunately, there were not enough related results to my paper to compare but there was enough material on “Mn-Zn ferrite” which we listed in a table and compared the magnetic properties and relaxivity values of them with our samples. We have compared the relaxivity properties of our samples with commercial contrast agents before.